# An Improved Generating Energy Prediction Method Based on Bi-LSTM and Attention Mechanism

Bo He [1], Runze Ma [2], Wenwei Zhang [2,3], Jun Zhu [4,*] and Xingyuan Zhang [1,*]

1    Department of Polymer Science and Engineering, University of Science and Technology of China, Hefei 230026, China; hebo01@zts.com.cn
2    Key Laboratory of Wireless Sensor Network and Communication of Chinese Academy of Sciences, Shanghai Institute of Microsystem and Information Technology, Shanghai 201899, China; marunze@mail.sim.ac.cn (R.M.); wenweizhang@mail.sim.ac.cn (W.Z.)
3    University of Chinese Academy of Sciences, Beijing 100049, China
4    Special Display and Imaging Technology Innovation Center of Anhui Province, Academy of Opto-Electric Technology, Hefei University of Technology, Hefei 230009, China
*    Correspondence: jzhu@hfut.edu.cn (J.Z.); zxym@ustc.edu.cn (X.Z.)

**Abstract:** The energy generated by a photovoltaic power station is affected by environmental factors, and the prediction of the generating energy would be helpful for power grid scheduling. Recently, many power generation prediction models (PGPM) based on machine learning have been proposed, but few existing methods use the attention mechanism to improve the prediction accuracy of generating energy. In the paper, a PGPM based on the Bi-LSTM model and attention mechanism was proposed. Firstly, the environmental factors with respect to the generating energy were selected through the Pearson coefficient, and then the principle and implementation of the proposed PGPM were detailed. Finally, the performance of the proposed PGPM was evaluated through an actual data set collected from a photovoltaic power station in Suzhou, China. The experimental results showed that the prediction error of proposed PGPM was only 8.6 kWh, and the fitting accuracy was more than 0.99, which is better than existing methods.

**Keywords:** Bi-LSTM; artificial neural networks; generating energy prediction

## 1. Introduction

The daily generating energy of a photovoltaic power station affects the power consumption of the local area [1–3], while the photovoltaic power generation has a relationship with environmental factors, such as sunshine duration, temperature, etc. Thus, the prediction of the generating energy helps the local power grid system to improve foreseeability and to create a proper generating schedule [4–7]. Since the main facility of a photovoltaic power station works outdoors, the environmental factors would affect the device's working state, making it meaningful to study this effect. For example, the characteristics of temperature changes on the quality of output current in solar power plants are studied in Indonesia [8]. In the global viewpoint, temperature and sunshine duration vary in different countries around the world, which makes the characteristics of solar plants generation different. It is a research focus to predict the generation based on environmental variation.

Generally, prediction is essentially a regression problem, the purpose of which is to build the relationship between environmental factors and generating energy. Hence, the machine learning-based methods have been widely used to achieve power generation prediction, such as outage forecasting, wind power prediction, stability forecasting, peak load prediction, etc.

The machine learning algorithm can treat big data efficiently [9], which can obtain the optimal parameters for PGPMs based on a lot of historical data, as well as make a prediction to generating energy through a trained model. Recently, the PGPMs based on

machine learning have been proposed for different types of power stations, such as wind power, thermal power, solar power, nuclear power, etc. Moreover, in order to achieve accurate prediction of daily generating energy of power stations, the input data set of existing PGPMs based on machine learning algorithm usually adopt all the environmental parameters that affect the power generation, which makes the computational complexity of such PGPM very high.

A PGPM based on support vector machine (SVM), one of the most commonly used algorithms in machine learning, was proposed in ref. [10], which applied an improved grid search method to optimize the parameters of C and g to improve the accuracy in forecasting wind power generation. The experimental results showed that the model was able to predict the real-time (15 min) wind power, and the accuracy was up to 78.49%. However, since the computational complexity is very high in scenarios with larger training samples, the SVM-based prediction model is only suitable for small-sample scenarios that can obtain the global optimization parameters.

In order to solve the limitations of the SVM-based PGPM in the large-sample condition, a lightweight PGPM based on ensemble decision tree haswas proposed in ref. [11], which can predict a power system's operating states in a real-time and in an on-line environment. In the proposed solution, an ensemble security predictor (ENSP) was developed and trained to predict and classify power system's dynamic operating states into secure, insecure, and intermediate transitional classes. Finally, the performance was evaluated with two different case studies performed on IEEE 118-bus and IEEE 300-bus test systems, and the experimental results showed that the prediction accuracy was up to 94.4%. However, in some circumstances, for the ensemble decision tree model, it is a challenge to find appropriate pruning schemes to remedy the decision tree due to the overfitting problem, which means the proposed model is only optimized for the existing data, namely, the proposed model is not quite suitable for unknown, new data.

Moreover, to improve the performance of the decision tree-based power generation prediction model, the random forest-based PGPM [12] is developed to forecast medium–long-term power load. In the proposed model, the total load is decomposed into the basic load affected by the economy and meteorological sensitive load affected by meteorological factors, and the prediction results are intelligently corrected by the wavelet neural network algorithm. The experimental results showed that the mean absolute percent error (MAPE) of the random forest-based PGPM was up to 1.43%, which is much better than decision tree-based model proposed in ref. [11]. However, the random forest-based model is equivalent to running multiple decision trees at the same time, which will inevitably have higher computational complexity than decision trees.

Apart from the above-mentioned statistical learning methods, the artificial neural network (ANN), which can simulate the human brain, has been widely used in the power generation prediction field in the recent years [13]. To improve the power production prediction for solar power stations, a PGPM based on the optimized and diversified artificial Neural Networks was proposed in ref. [14]. The method is optimized in terms of the number of hidden neurons and improved in terms of diverse training datasets used to build ANN. The simulation results showed that the proposed approach outperformed three benchmark models, with a performance gain reaching up to 11% for RMSE (root-mean-square error) metric, and the confidence level reaches up to 84%. However, such methods employ classical neural networks, which may not be suitable for some time-varying sequence data of environmental factors.

Generally, for time-varying sequence data, the model based on recurrent neural network (RNN) can provide higher prediction accuracy [15]. The Long Short-Term Memory (LSTM) [16], an improved RNN, could solve the problems of gradient disappearance and gradient explosion when training long sequence data in RNN, making it superior in time sequence forecasting problems [17]. The LSTM network has a strong memory function, which can establish the correlation between the data before and after, thereby improving the prediction accuracy. Based on the above advantages of LSTM, a PGPM based on the

high-performance K-Means-long-short-term-memory (K-Means-LSTM) was proposed to predict the power point of wind power in ref. [18], and the simulation results showed that the prediction error (RMSE) of the proposed PGPM reached 62 kW, achieving higher accuracy than RNN-based methods.

However, the LSTM-based PGPM can only capture the data features of the former part of the time sequence, which in turn leads to very limited performance of such methods in some scenarios. As an improved version of LSTM, the Bidirectional LSTM (Bi-LSTM) has better performance via adding a reverse-calculation module. Hence, a Bi-LSTM-based PGPM, which is used to predict the abnormal electricity consumption in power grids, was proposed in [19]. In the Bi-LSTM-based PGPM, the framework of Tensorflow was used to achieve feature extraction and power generation prediction. Final experimental results showed that the accuracy of the Bi-LSTM-based PGPM reached up to 96.1%, which is better than that of the LSTM-based PGPM proposed in ref. [18] (94.5%).

Generally, the Bi-LSTM model can enhance the mining of correlation information of time series feature to some extent; however, it can only extract local features, and it is difficult to obtain global correlation, resulting in the loss of feature correlation information. Simultaneously, such a model only focuses on the inherent relationship between the input features and the target feature, so the input features of each time are assigned the same weight. Nevertheless, the correlation between the input and target characteristics of electricity consumption varies with time, which puts forward higher requirements for the mining of time series correlation of input features.

Hence, in order to improve the performance of PGPMs based on Bi-LSTM, an Attention-Bi-LSTM PGPM based on attention mechanism and Bi-LSTM is proposed in this paper, which adequately employs the advantages of the attention mechanism and Bi-LSTM network. The main contribution of this paper is the way in which the attention mechanism is introduced. To solve this, appropriate attention layers have to be selected and designed to efficiently utilize historical data.

Moreover, existing machine learning-based PGPMs usually use all environmental parameters that affect power generation as input data sets, which can inevitably increase the computational burden of computers. In order to improve computational efficiency, the feature selection algorithm based on Pearson correlation theory [20] is proposed before constructing the proposed PGPM.

The remaining of this paper is organized as follows. Section 2 details the principle of environmental factors selection method based on Pearson coefficient theory. Section 3 presents the methodology of the prediction method. Section 4 elaborates data processing procedures. Section 5 shows experimental layout and relative results. Section 6 concludes the paper and looks forward to future work.

## 2. Feature Selection

According to the previous analysis, the daily generating energy is related to environmental factors for photovoltaic power stations, and there are correlations between the above-mentioned environmental factors. Therefore, finding the correlation between various environmental factors and selecting appropriate environmental factors as the input dataset can inevitably reduce the computational complexity of prediction models.

Generally, the environmental factors such as daily average temperature, maximum temperature, minimum temperature, daily sunshine duration, average cloud cover, average humidity, minimum humidity, precipitation from 8:00 a.m. to 8:00 p.m., etc., can affect power generation. Under normal circumstances, the more environmental factors, the larger the processing of high-dimensional vectors, as these factors would constitute the input feature vector, and the complexity of calculations will be improved greatly. To reduce the calculation complexity, these environmental factors should be properly selected, and the Pearson correlation coefficients that can evaluate the correlation between environmental factors and generating energy are introduced into the paper.

Pearson correlation coefficient is a value between −1 and 1 that denotes the similar trend between two datasets. For two random variables $X$ and $Y$, the Pearson correlation coefficient can be expressed by:

$$\rho_{XY} = \frac{cov(X,Y)}{r_X r_Y} = \frac{E(XY) - E(X)E(Y)}{\sqrt{E(X^2) - E^2(X)}\sqrt{E(Y^2) - E^2(Y)}} \tag{1}$$

where $cov(X,Y)$ means the covariance between $X$ and $Y$; $\rho_X$ and $\rho_Y$ are the standard deviation of $X$ and $Y$ respectively; $E(.)$ function means the random variable's expectation.

In the paper, the Pearson correlation coefficient between environmental factors and generating energy can be calculated by:

$$r = \frac{N\sum x_i y_i - \sum x_i \sum y_i}{\sqrt{N\sum x_i^2 - (\sum x_i)^2}\sqrt{N\sum y_i^2 - (\sum y_i)^2}} \tag{2}$$

where $r$ is the Pearson coefficient; $x_i$ and $y_i$ are the environmental factors and corresponding generating energy respectively; $N$ is the amount of historical data samples.

Hence, in order to select the optimal environmental factors to construct the input dataset, the Pearson coefficients between environmental factors and generating energy obtained from a photovoltaic power station in Suzhou, China (Supplementary Materials), were used and the results are shown in Table 1.

**Table 1.** Pearson coefficients between environmental factors and generating energy.

| Environmental Factors | Pearson Coefficient |
|---|---|
| Daily average temperature | 0.42551 |
| Maximum temperature | 0.54173 |
| Minimum temperature | 0.27529 |
| Average humidity | −0.69062 |
| Minimum humidity | −0.74763 |
| Precipitation from 8:00 a.m. to 8:00 p.m. | −0.33582 |
| Daily sunshine duration | 0.83609 |
| Average cloud cover | −0.59997 |

According to Pearson coefficient theory, factors with positive Pearson coefficients have good correlation with the generating energy, which means they are suitable to be regarded as the input data features to predict the generating energy. As can been found in Table 1, some factors such as average humidity, minimum humidity, precipitation from 8:00 a.m. to 8:00 p.m., and average cloud cover could be filtered because they have a weak correlation with generating energy. Hence, the remaining four environmental factors are taken to compose the input feature vector, which means the data feature vectors are four-dimensional.

## 3. The Methodology

### 3.1. Bi-LSTM Model

Generally, Bi-LSTM is composed by two LSTM models of the forward and backward direction, which can capture long-term dependencies in one direction. Hence, the Bi-LSTM allows more information to be preserved by capturing long-term dependencies in both directions, which is suitable for power generation forecasting scenarios that require big data processing. The architecture of Bi-LSTM model can be shown as Figure 1.

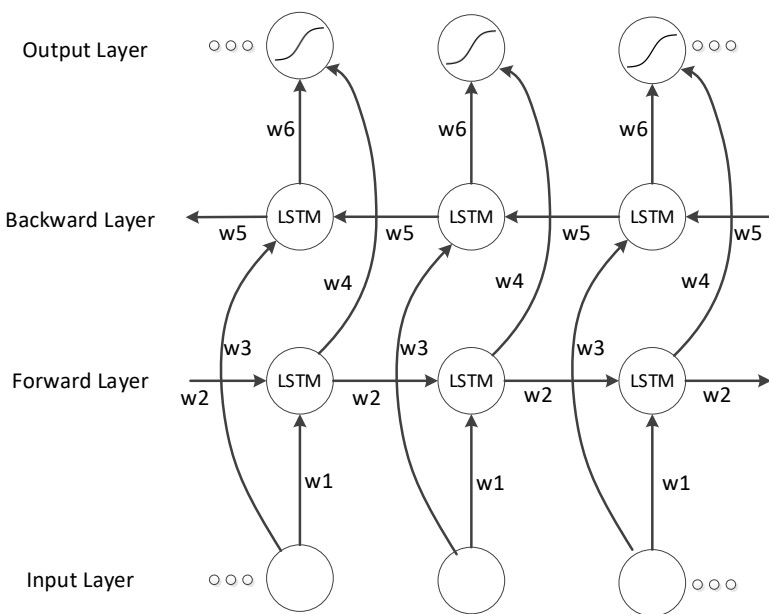

**Figure 1.** Architecture of the Bi-LSTM model.

From Figure 1, it can be found that the forward directional LSTM is used to produce the past information of input sequences, while the backward directional LSTM can gain the future information of input sequences. Finally, the final output is obtained by combining the corresponding time output of forward directional LSTM and backward directional LSTM at each time, which can be expressed by:

$$h_t = f(w_1 x_t + w_2 h_{t-1}) \tag{3}$$

$$h'_t = f(w_3 x_t + w_5 h'_{t+1}) \tag{4}$$

$$o_t = g(w_4 h_t + w_6 h'_t) \tag{5}$$

where $h_t$ and $h'_t$ are current node outputs of the forward and backward direction respectively; $o_t$ is the output of current cell; $w_1$, $w_2$, $w_3$, $w_4$, $w_5$ and $w_6$ are the weight coefficients.

According to Equations (3)–(5), $w_1$ and $w_3$ are the weights of the input to the forward and backward hidden layers, $w_2$ and $w_5$ are the weights between the same hidden layers, while $w_4$ and $w_6$ are the weights of the forward and backward hidden layers to the output layers. Compared with LSTM, Bi-LSTM improves the globality and integrity of feature extraction.

### 3.2. Feature Attention Mechanism

Generally, the feature attention mechanism can improve the performance of Bi-LSTM by dynamically assigning the attention weight to input features, as well as the correlation between hidden layer and target features being mined, which can effectively reduce the loss of feature correlations. The architecture of the feature attention mechanism is shown in Figure 2.

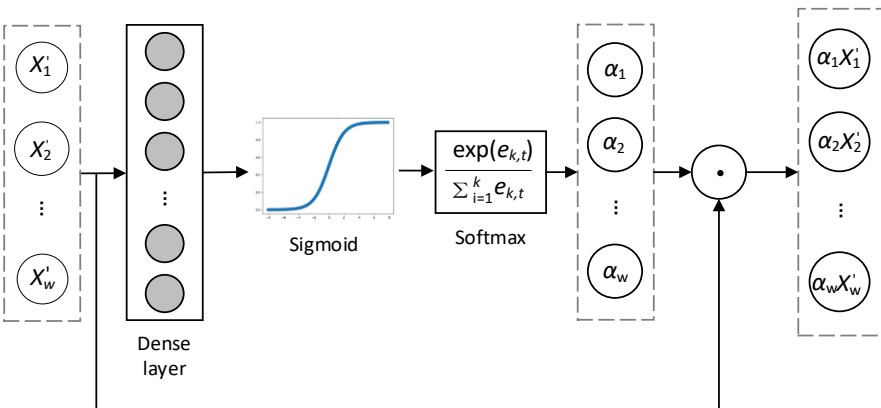

**Figure 2.** Architecture of feature attention mechanism.

From Figure 2, the input feature vector of time sequences with *K* hidden layer features can be described as $X_t = [X_{1,t}, X_{2,t}, \ldots, X_{k,t}]$. Then, a single layer neural network is used to calculate the attention weight vector, which can be expressed by:

$$e_{k,t} = \sigma(W_e X_t + b_e) \tag{6}$$

where *t* is the time length of input sequences depending on sampling rates, and $e_{k,t} = [e_{1,t}, e_{2,t}, \ldots, e_{k,t}]$ is regarded as the combination of attention weight coefficients corresponding to the input characteristics of current moments. $W_e$ is the trainable weight matrix, $b_e$ is an offset vector, and $\sigma(.)$ is a sigmoid activation function.

The data sequence generated by the sigmoid activation function is normalized by the softmax function, which is denoted as:

$$\alpha_{k,t} = \frac{\exp(e_{k,t})}{\sum\limits_{i=1}^{k} e_{i,t}} \tag{7}$$

where $\alpha_{k,t}$ is the attention weight of character *k*, and the resulting attention weight $\alpha_{k,t}$ and hidden layer feature vector $X_t'$ are recalculated as a weighted feature vector $X_{a\_t}'$, which can be expressed by:

$$X_{a\_t}' = a_t \odot X_t' = [a_{1,t}x_{1,t}, a_{2,t}x_{2,t}, \cdots, a_{k,t}x_{k,t}] \tag{8}$$

### 3.3. Temporal Attention Mechanism

Apart from the feature attention mechanism, the temporal attention mechanism can allocate attention weight to the temporal information carried by each historical moment of the input sequence to distinguish its influence on the output of the current time. At the same time, the time sequence of each historical moment can be extracted independently and the information expression of the critical moment can be enhanced; the architecture of the temporal attention mechanism is shown in Figure 3.

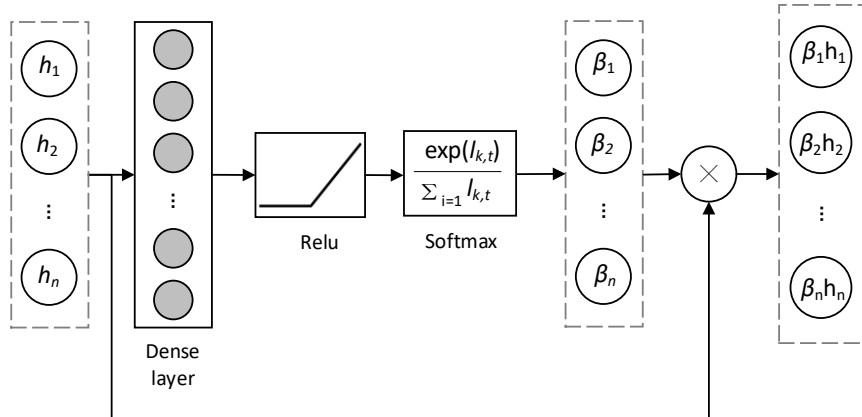

**Figure 3.** Architecture of the temporal attention mechanism.

From Figure 3, it can be found that the input is the hidden layer state of the Bi-LSTM network iterated to time, which can be expressed by $h_t = [h_{1,t}, h_{2,t}, \ldots, h_{n,t}]$, where $n$ is the time window length of input sequences. The temporal attention weight vector $l_t$ of the current moment corresponding to each historical moment can be described as:

$$l_t = \mathrm{Re}LU(W_d X_t + b_d) \tag{9}$$

where $l_t = [l_{1,t}, l_{2,t}, \ldots, l_{k,t}]$; $W_d$ is a trainable weight matrix; $b_d$ is a bias vector; and $\mathrm{Re}LU(.)$ is an activation function to increase feature differences and make the weight distribution more centralized.

Moreover, from Figure 3, it can be seen that the input sequence generated by the activation function is normalized by the softmax function to obtain the temporal attention weight, which can be expressed by $\beta_t = [\beta_{1,t}, \beta_{2,t}, \ldots, \beta_{k,t}]$, where $\beta_{k,t}$ is the attention weight of character $k$, which can be denoted as:

$$\beta_{k,t} = \frac{\exp(l_{k,t})}{\sum\limits_{i=1}^{k} l_{i,t}} \tag{10}$$

Hence, the weighted feature vector $h_t'$ can be recalculated via data feature vector $h_t$ generated by the hidden layer, which can be expressed by:

$$h_t' = \beta_t \otimes h_t = \sum_{i=1}^{k} \beta_{i,t} h_{i,t} \tag{11}$$

### 3.4. The Proposed Attention-Bi-LSTM PGPM

In the paper, the Attention-Bi-LSTM PGPM based on the attention mechanism and Bi-LSTM network is proposed, which consists of an input layer, feature attention layer, Bi-LSTM layer, temporal attention layer, residual connected layer, and fully connected layer, and the architecture of the Attention-Bi-LSTM PGPM is shown in Figure 4.

From Figure 4, it can be found that a Bi-LSTM network is built to extract the hidden temporal correlation information from the input sample $X_t$, which is composed of the history sequence and related four-dimensional input feature vector extracted from environmental factors. The sample is fed into first Bi-LSTM network and the hidden layer feature $X_t'$ is obtained.

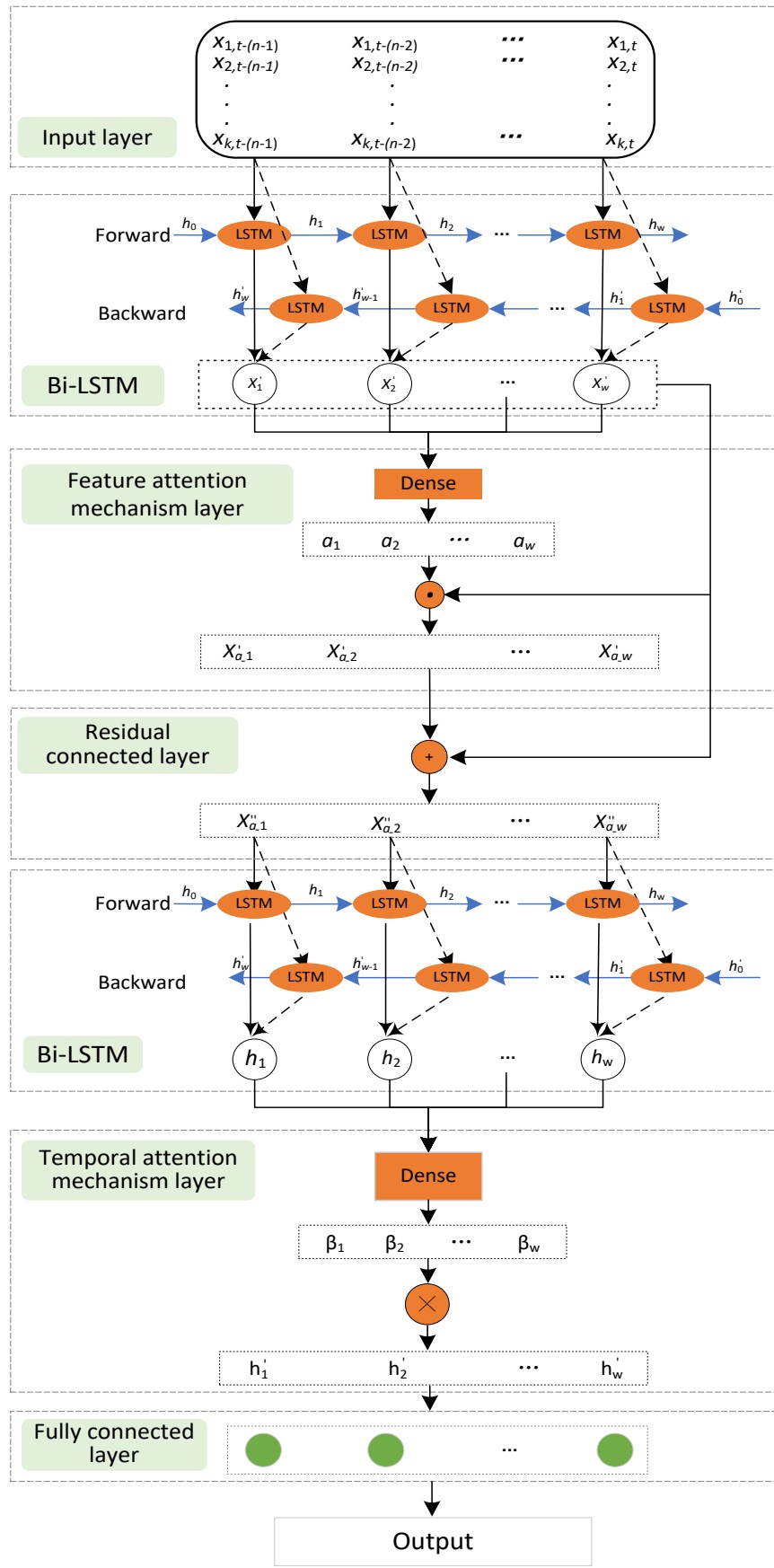

**Figure 4.** Architecture of Attention-Bi-LSTM PGPM.

Then, the feature attention mechanism was used to explore the potential correlation between hidden layer features $X_t'$. The features $X_t'$ extracted from the first Bi-LSTM were sent to the feature attention layer. In order to extract the hidden layer features $X_t'$, the attention weight of features was allocated dynamically. Based on the above statements, the weighted hidden layer feature $X_{a\_t}'$ was obtained by dynamic distribution of the feature attention weight.

Next, the weighted feature $X_{a\_t}'$ was residually linked to the original feature $X_t'$, which was fed into the second Bi-LSTM to obtain the hidden layer feature $h_t$. Moreover, the correlation between the historical sequence and the feature $h_t$ was mined in the second Bi-LSTM's hidden layer, as well as the weighted feature vector $h_t'$ being mined in the temporal attention layer. Finally, the power generation was predicted in the fully connected layer with the above-mentioned parameters.

## 4. Data Processing

### 4.1. Data Cleaning

In this paper, a historical dataset collected from a photovoltaic power station with a sampling rate of 1 day, which includes daily average temperature, maximum temperature, minimum temperature, daily sunshine duration, and daily generating energy, was introduced into the experiment [SM]. The input data sample is a 4-dimensional vector, which denotes the above-mentioned four environmental features, and every input feature vector corresponds to a daily generating energy, as the output value.

For data cleaning, firstly, the data sample with missing or invalid features was preprocessed. In this paper, the data sample with invalid features was eliminated directly.

Secondly, different features have values of different ranges, making it necessary to normalize the feature data. The normalized value could be calculated by:

$$
\begin{cases}
\overline{x} = \frac{1}{n} \sum\limits_{i=1}^{n} x_i \\
std(x) = \sqrt{\frac{1}{n} \sum\limits_{i=1}^{n} (x_i - \overline{x})^2} \\
y_i = \frac{x_i - \overline{x}}{std(x)}
\end{cases}
\tag{12}
$$

where $x_i$ is the $i$-th original feature value; $y_i$ is the $i$-th normalized feature value; $n$ is the amount of data samples.

### 4.2. Division of Dataset

To train the prediction model parameters, which are mainly some structural weight values, 75% of historical data samples were recognized as the training dataset, and the remaining 25% of data samples were taken as the testing dataset to examine the prediction efficiency. The ensemble division of dataset is shown in Figure 5.

As shown in Figure 5, the training process adopts a cross validation mechanism, composed by many epochs. In each epoch, 90% of the training samples are regarded as a sub-training set, and the remaining 10% of the training samples are regarded as the sub-testing dataset. The partition scheme of the sub-training dataset and sub-testing dataset is to divide them randomly. From Figure 5, it can be found that the optimal parameters are obtained through multiple cross-validation, which was used to provide a basis for the subsequent experiments.

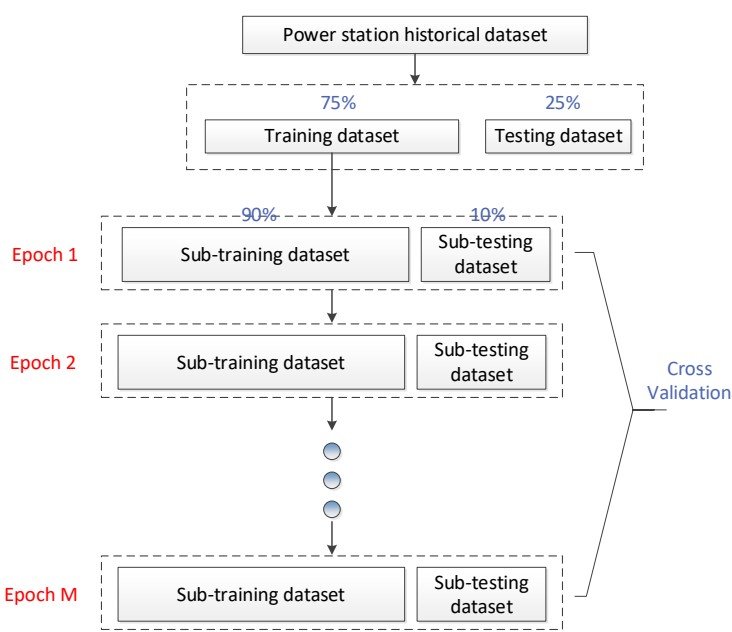

**Figure 5.** Division of the dataset.

## 5. Experimental Results and Analysis

*5.1. Parameter Tuning and Statistical Analysis*

In order to illustrate the advantages of the proposed PGPM, the performance of existing PGPMs based on Support Vector Regression (SVR) [21], Decision Tree [22], Random Forest [23], LSTM [24], and Bi-LSTM [25] were compared with the Attention-Bi-LSTM PGPM proposed in the paper, and the main experimental parameters of PGPMs based on SVR, Decision Tree, and Random Forest were tuned, as shown in Tables 2–4, respectively.

**Table 2.** Parameter tuning of PGPM based on SVR.

| Penalty C | RBF Gamma | Prediction Error (kWh) |
|---|---|---|
| 100 | 1 | 238.9 |
| 1 | 0.01 | 479.3 |
| 0.1 | 0.01 | 489.1 |

**Table 3.** Parameter tuning of PGPM based on Decision Tree.

| Max Depth | Prediction Error (kWh) |
|---|---|
| 4 | 255.9 |
| 5 | 243.6 |
| 6 | 236.0 |
| 10 | 291.7 |
| 90 | 305.6 |

**Table 4.** Parameter tuning of PGPM based on Random Forest.

| Number of Estimators | Minimum Samples of Subtree | Minimum Samples of Leaf | Prediction Error (kWh) |
|---|---|---|---|
| 200 | 2 | 1 | 231.8 |
| 200 | 2 | 4 | 232.1 |
| 100 | 2 | 1 | 232.9 |
| 400 | 4 | 1 | 232.9 |
| 400 | 4 | 2 | 232.8 |

From Tables 2–4, the best parameters of each algorithm could be determined, for the best prediction accuracy was achieved.

Moreover, the essence of proposed Attention-Bi-LSTM PGPM is an improved version of PGPMs based on LSTM and Bi-LSTM. In order to ensure the comparability and accuracy of subsequent experimental results, the experiments parameters of the above three LSTM-based PGPMs are the same in the paper, and the related parameters are shown in Table 5.

**Table 5.** Related parameters of LSTM-based PGPMs.

| Category | Parameter |
| --- | --- |
| Length of Time Sequence | 4 |
| Bi-LSTM Hidden Layer Neurons | 350 |
| Learning Rate | 0.01 |
| Batch Size | 64 |
| Optimization Algorithm | Adam |
| Loss Function | Mean Squared Error (MSE) |
| Neuron Loss Rate | 0.1 |

Furthermore, the statistical analysis was performed for the selected parameter configurations, the way of which is to run the model training and prediction 50 times. Each time, the training dataset and testing dataset were partitioned randomly to evaluate the statistical stability of these models, and the results are shown in Table 6.

**Table 6.** Statistical analysis on the studied methods.

| Method | Average of RMSE (kWh) | Standard Deviation of RMSE (kWh) |
| --- | --- | --- |
| SVR | 238.9 | 2.3 |
| Decision Tree | 236.0 | 2.7 |
| Random Forest | 231.8 | 1.9 |
| LSTM | 29.7 | 1.5 |
| Bi-LSTM | 18.3 | 1.8 |
| Attention-Bi-LSTM (Ours) | 8.6 | 1.2 |

Table 6 shows the standard deviation for each algorithm is only 1~2 kWh, which means the prediction result is stable when the parameters are determined. Therefore, the subsequent comparison of parameter-dependent results could reflect the performance gaps of different methods from the statistical viewpoint.

Moreover, in order to evaluate the performance of the above algorithms, the Python scikit-learn library was employed to implement the PGPMs based on SVR, Decision Tree, and Random Forest algorithms, while the Tensorflow library was employed to implement the PGPMs based on LSTM, Bi-LSTM, and the proposed Attention-Bi-LSTM.

*5.2. Experimental Results*

According to above-mentioned relevant experimental parameters shown in Table 2 to Table 5 and experimental layouts, the visualized experimental results within half a year output by six PGPMs mentioned above are shown in Figure 6.

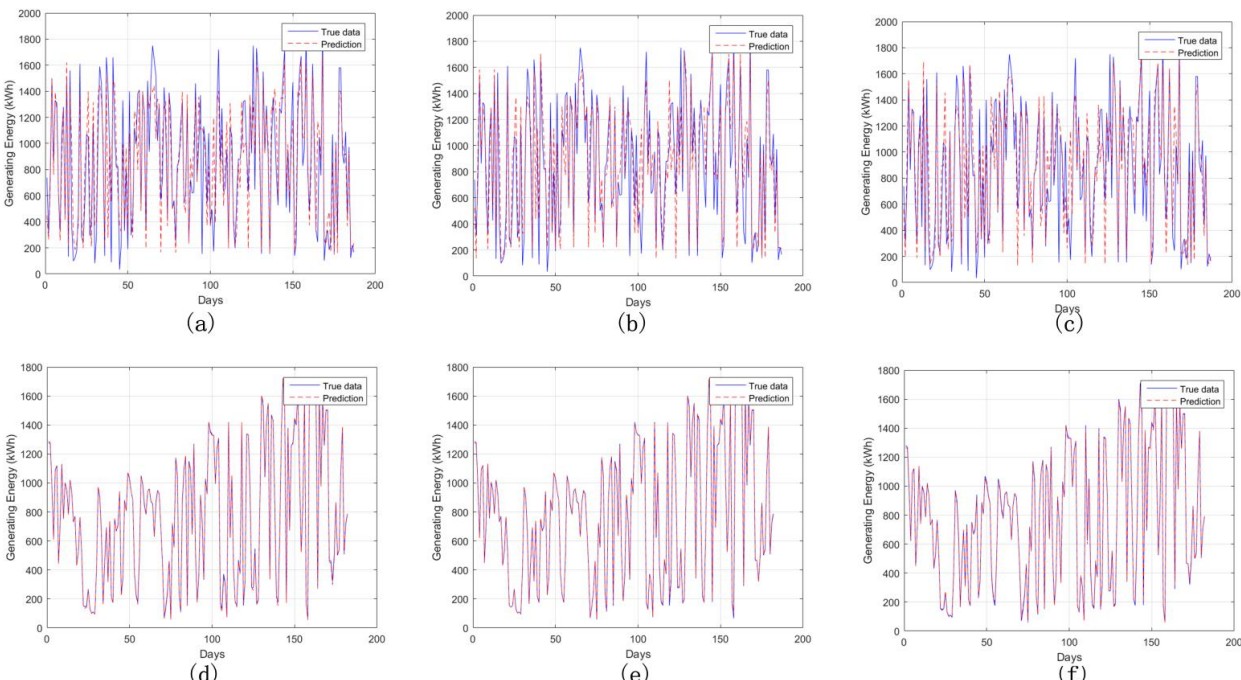

**Figure 6.** Prediction results of different algorithms. (**a**) PGPM based on SVR; (**b**) PGPM based on Decision Tree; (**c**) PGPM based on Random Forest; (**d**) PGPM based on LSTM; (**e**) PGPM based on Bi-LSTM; (**f**) Ours.

As shown in Figure 6, it can be found that the deviations between the true data and experimental results of PGPMs based on SVR, Decision Tree, and Random Forest were more obvious than that generated of PGPMs based on LSTM, Bi-LSTM, and Attention-Bi-LSTM. Summarily, the LSTM-based PGPMs are very suitable for power generation forecasting scenarios. However, according to Figure 6d–f, it can be seen that from the visualization point of view, the performance of Attention-Bi-LSTM PGPM proposed in this paper is basically the same as that of the other LSTM-based PGPMs. Therefore, to further illustrate the advantages of the proposed PGPM, this paper evaluates the performance of above-mentioned PGPMs from a quantitative perspective.

Besides, in the training procedure of the proposed PGPM, the model converges very quickly, as presented in Figure 7.

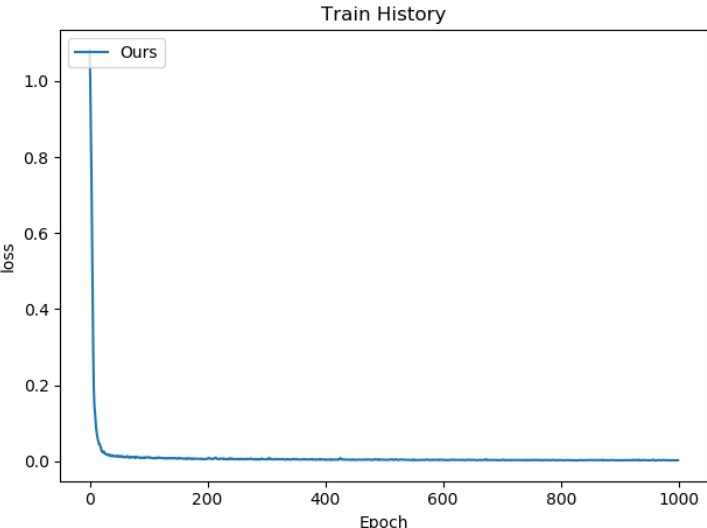

**Figure 7.** Convergence curve of the proposed PGPM.

As Figure 7 shows, the loss function of the model decreased quickly, and converged nearly to zero within the first 100 epochs, which means in the practical training procedure it could be finished very fast.

### 5.3. The Quantitative Comparison of Results

To evaluate the performances of above-mentioned PGPMs more precisely, Mean Absolute Error, Root of Mean Square Error (RMSE), and Mean Absolute Percentage Error (MAPE) of each PGPM were evaluated and compared. Moreover, a R-square coefficient [26] is also introduced into the paper to calculate the fitting accuracy, which can be expressed by

$$MAE = \frac{1}{n}\sum_{i=1}^{n} |\hat{y}_i - y_i| \tag{13}$$

$$RMSE = \sqrt{\frac{1}{n}\sum_{i=1}^{n} (\hat{y}_i - y_i)^2} \tag{14}$$

$$MAPE = \frac{1}{n}\sum_{i=1}^{n} \left| \frac{\hat{y}_i - y_i}{y_i} \right| \times 100\% \tag{15}$$

$$R - square = \frac{\sum_{i=1}^{n} (\hat{y}_i - \bar{y}_i)^2}{\sum_{i=1}^{n} (y_i - \bar{y}_i)^2} \tag{16}$$

where $y_i$ is the generating energy (true data) of the $i$-th sample; $\hat{y}_i$ is the prediction of the $i$-th sample; $R - square$ is a coefficient with a range of [0 1], and the closer this value is to 1, the higher the fitting accuracy.

According to Equations (13) to (16), the prediction errors and fitting accuracy of above-mentioned PGPMs are shown in Table 7.

**Table 7.** Comparison of different PGPMs.

| Method | MAE (kWh) | RMSE (kWh) | MAPE (%) | Fitting Accuracy (R-Square) |
|---|---|---|---|---|
| SVR | 166.7 | 238.9 | 40.7 | 0.7617 |
| Decision Tree | 160.3 | 236.0 | 37.9 | 0.7675 |
| Random Forest | 160.6 | 231.8 | 38.8 | 0.7591 |
| LSTM | 25.5 | 29.7 | 5.7 | 0.9959 |
| Bi-LSTM | 13.7 | 18.3 | 3.6 | 0.9984 |
| Ours | 10.2 | 8.6 | 2.8 | 0.9997 |

As Table 7 shows, the prediction errors of the proposed PGPM were 10.2 kWh, 8.6 kWh, and 2.8%, which were the smallest among these six algorithms. Moreover, from Table 6, taking RMSE as an example, it can be found that the prediction errors of the PGPMs based on SVR, Decision Tree, and Random Forest were 238.9 kWh, 236.0 kWh, and 231.8 kWh, respectively, which are generally more than 200 kWh, as well as that of the PGPMs based on LSTM and Bi-LSTM being less than 30 kWh. Hence, the performances of LSTM- and Bi-LSTM-based PGPM are better than that of SVR-, Decision Tree-, and Random Forest-based PGPMs. Simultaneously, with the introduction of the attention mechanism, the proposed PGPM also achieved better prediction accuracy than that of LSTM- and Bi-LSTM-based PGPMs. The metrics of MAE and MAPE showed similar results.

Additionally, the fitting accuracy was also evaluated in this paper. Fitting accuracy is another indicator for evaluating prediction efficiency, which represents the relative prediction error and can be used as a sign of the similarity between the predicted value and the true value. From Table 6, it can be found that the fitting accuracy of the proposed PGPM was 0.9997, slightly more than that based on LSTM and Bi-LSTM, but obviously more than that of SVR-, Decision Tree-, and Random Forest-based PGPMs. Therefore, in the metric of

fitting accuracy, the proposed Attention-Bi-LSTM PGPM achieves the best performance, and is consequently very suitable for application in power generation forecasting scenarios.

*5.4. Comparison of Multi-Step Prediction Results*

Moreover, in order to evaluate the influence of proposed PGPM with the input time sequences of various step lengths, an experiment was also implemented based on different time steps, and the experimental results are shown in Table 7.

From Table 8, it can be found that there was a positive correlation between the prediction error and step size; in other words, the prediction error increased with respect to step length increases. Synchronously, the fitting accuracy had a negative correlation with step length, that is, the fitting accuracy decreased as the step length increased. The reason for the above phenomenon is that the dependence between the power generation and time sequences is weakened with the increase of step length. In summary, when the time step of input time sequences is four, the PGPM proposed in this paper can meet the demand for power generation forecasting.

**Table 8.** Comparison of multi-step prediction results.

| Time Step | Evaluation Criteria | |
|:---:|:---:|:---:|
| | Prediction Error (kWh) | Fitting Accuracy (R-Square) |
| 4 | 8.6408 | 0.9997 |
| 8 | 15.2754 | 0.9989 |
| 10 | 18.0235 | 0.9985 |
| 14 | 23.8192 | 0.9974 |

## 6. Conclusions

The contribution of this paper was to propose a generating energy prediction model based on the attention mechanism and Bi-LSTM, which improve the prediction accuracy, and the experimental results showed that the performance of the proposed PGPM is much better than that of PGPMs based on SVR, Decision Tree, Random Forest, LSTM, and Bi-LSTM. The challenge of this work was how to employ attention mechanism efficiently. To solve this, feature attention layer and temporal attention layer were introduced to enhance the prediction performance, because these attention layers could help the algorithm to utilize the most important features and the most critical moments.

Moreover, compared with the existing PGPMs, this paper mines the correlation of environmental factors that affect photovoltaic power generation before implementing the proposed PGPM, and thereby the calculation efficiency can be improved by eliminating environmental factors that are weakly related to power generation.

However, the data features of the proposed PGPM are few, and only the meteorological factors are considered as the input source. In the future, to further optimize the accuracy of the prediction method, other data features can be introduced to construct a more accurate input source.

**Supplementary Materials:** The following supporting information can be downloaded at: https://www.mdpi.com/article/10.3390/electronics11121885/s1; also in ftp://simitPublic:Simit123@47.116.99.105 (accessed on 12 June 2022), the generation data from a power station in Suzhou, China.

**Author Contributions:** Conceptualization, B.H., R.M. and X.Z.; data curation, W.Z.; formal analysis, R.M. and W.Z.; investigation, B.H.; methodology, R.M. and W.Z.; project administration, B.H.; resources, B.H.; software, W.Z.; supervision, J.Z.; validation, J.Z. and X.Z.; visualization, W.Z.; writing—original draft, R.M. and W.Z.; writing—review and editing, B.H. and R.M. All authors have read and agreed to the published version of the manuscript.

**Funding:** This research received no external funding.

**Institutional Review Board Statement:** Not applicable.

**Informed Consent Statement:** Not applicable.

**Data Availability Statement:** Not applicable.

**Acknowledgments:** Thanks are due to Wujiang photovoltaic power station for assistance with generation and environmental data.

**Conflicts of Interest:** The authors declare no conflict of interest.

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
