# Peer review of "An Improved Generating Energy Prediction Method Based on Bi-LSTM and Attention Mechanism"

_electronics, doi:10.3390/electronics11121885_

Round 1

Reviewer 1 Report

electronics-1709771

Although the study is promising, there are some concerns regarding the details of the presented study as well as the organization of the paper. Furthermore, the structure of the submitted manuscript and the literature survey requires some improvements.

Since several papers are published in the field of Energy Prediction methods, the reviewer is curious to see the attempts of the authors in revising the paper and answering the questions about the proposed study. The reviewer has the following suggestions/comments:

  1. How many years of data were taken?
  2. It is mentioned in line 133 that “Generally, the environmental factors such as daily average temperature, maximum temperature, minimum temperature, daily sunshine duration, average cloud cover”. What is the effect of other parameters on this?
  3. What is the effect of wind speed and wind direction? Why these parameters are not included as features?
  4. What will be the difference in accuracy if you predict the radiation from short-term to long-term?
  5. Elaborate on the data and its cleaning as a separate section.
  6. Briefly explain why data are taken from 8 am to 20 pm?
  7. Due to some alignment issues, some references are showing “Error! Reference source not found.”. Correct it
  8. Include a few lines about the challenges faced in this work in the conclusion.

Reviewer 2 Report

  1. Throughout the paper, it is written Error! Reference not found. The reviewer did not find the relation between this statement and the presented results.
  2. The prediction error is 6.2 kWh, this is not clear. The percentage representation would be better to read.
  3. The energy production by the PV depends on the environmental effects which has an impact on prediction accuracy. Although the authors has indicated this fact, however, there is no analysis about the environmental impact on the selected and studied system. A short analysis can be added to see how the prediction accuracy varies at different level of environmental variation, such as 5 or 10% increase in temperature etc.

Also, the introduction section can be improved by adding the environmental impacts on PV power generations around the world. An example research (TEMPERATURE EFFECT ON PV OUTPUT POWER VARIABILITY)  can be found to gain experience of PV energy variation due to temperature in the different countries.  

  1. The proposed method shows an improved result. However, what are the sciences behind of improving the prediction results are need to be clarified.
  2. The contribution of this study to be added at the last section of the paper.

Reviewer 3 Report

The paper topic is of interest due the selection of employing the AI to renewable energy source. The technique used providing very good results with significant accuracy. The paper needs to add few recent references from 2022, to raise the value of the article.

Reviewer 4 Report

I have the following comments on the manuscript.

  1. The contribution should be further clarified after identifying research gaps
  2. The prediction errors should be presented in MAE, RMSE and MAPE instead of prediction errors
  3. Statistical analysis of the forecasting model is crucial to demonstrate the effectiveness of the model. The current results do not reflect the model's effectiveness
  4. Convergence curves of all the forecasting models should be presented to determine the best model and the reasons for that need to be described.
  5. More forecasting results and comparisons are needed before claiming the best model.
  6. Parameter tuning of all the algorithms should be carried out and how you have done should be described. This is very important for performance improvement.
  7. Correct the lines "in Error! Reference source not found.." 

Round 2

Reviewer 1 Report

Paper can be accepted in present form 

Author Response

Point: Paper can be accepted in present form

Response: Thank you very much for the approval to the paper

Reviewer 2 Report

Still, the paper is full of errors with "Error! Reference source not found., 188 Error! Reference source not found. and Error! Reference source not found". This indicates that the revision was done carelessly. 

It is good to indicate the temperature effect in the local context, however, the analysis of this variation needs to be present in the global context, which is not been considered. 

Reviewer 4 Report

Unfortunately, the authors did not address any of the previous concerns, indicating the authors' lack of expertise in the areas.

Round 3

Reviewer 2 Report

No further comments. 

Author Response

Thank you very much.

Reviewer 4 Report

The authors must carry out a Statistical analysis to determine the effectiveness of the proposed method as the outcome of the method varies in each run. Moreover, this method is neither new nor any new results concluded. The response is not satisfactory: "Point 3: Statistical analysis of the forecasting model is crucial to demonstrate the effectiveness of the model. The current results do not reflect the model's effectiveness".

The following previous comments were not addressed.

  1. The contribution should be further clarified after identifying research gaps
  2. Statistical analysis of the forecasting model is crucial to demonstrate the effectiveness of the model. The current results do not reflect the model's effectiveness
  3. Convergence curves of all the forecasting models should be presented to determine the best model and the reasons for that need to be described.
  4. Parameter tuning of all the algorithms should be carried out and how you have done should be described. This is very important for performance improvement.
